# The Image Local Autoregressive Transformer

**Chenjie Cao, Yuxin Hong, Xiang Li, Chengrong Wang, Chengming Xu, Yanwei Fu,** *Xiangyang Xue
School of Data Science
Fudan University
{20110980001,yanweifu}@fudan.edu.cn

## Abstract

Recently, AutoRegressive (AR) models for the whole image generation empowered
by transformers have achieved comparable or even better performance compared
to Generative Adversarial Networks (GANs). Unfortunately, directly applying
such AR models to edit/change local image regions, may suffer from the problems
of missing global information, slow inference speed, and information leakage of
local guidance. To address these limitations, we propose a novel model – image
Local Autoregressive Transformer (iLAT), to better facilitate the *locally guided
image synthesis*. Our iLAT learns the novel local discrete representations, by the
newly proposed local autoregressive (LA) transformer of the attention mask and
convolution mechanism. Thus iLAT can efficiently synthesize the local image
regions by key guidance information. Our iLAT is evaluated on various locally
guided image syntheses, such as pose-guided person image synthesis and face
editing. Both quantitative and qualitative results show the efficacy of our model.

## 1 Introduction

Generating realistic images has been attracting ubiquitous research attention of the community for
a long time. In particular, those image synthesis tasks involving persons or portrait [6, 28, 29]
can be applied in a wide variety of scenarios, such as advertising, games, and motion capture, etc.
Most real-world image synthesis tasks only involve the local generation, which means generating
pixels in certain regions, while maintaining the semantic consistency, *e.g.*, face editing [19, 1, 40],
pose guiding [36, 55, 47], and image inpainting [51, 30, 49, 53]. Unfortunately, most works can
only handle the well aligned images of 'icon-view' foregrounds, rather than the image synthesis of
'non-iconic view' foregrounds [47, 24], *i.e.*, person instances with arbitrary poses in cluttered scenes,
which is concerned in this paper. Even worse, the global semantics tend to be distorted during the
generation of previous methods, even if subtle modifications are applied to a local image region.
Critically, given the local editing/guidance such as sketches of faces, or skeleton of bodies in the first
column of Fig. 1(A), it is imperative to design our new algorithm for *locally guided image synthesis*.

Generally, several inherent problems exist in previous works for such a task. For example, despite
impressive quality of images are generated, GANs/Autoencoder(AE)-based methods [51, 47, 19, 30,
18] are inclined to synthesize blurry local regions, as in Fig. 1(A)-row(c). Furthermore, some inspiring
autoregressive (AR) methods, such as PixelCNN [32, 41, 23] and recent transformers [8, 14], should
efficiently model the joint image distribution (even in very complex background [32]) for whole
image generation as Fig. 1(B)-row(b). These AR models, however, are still not ready for locally
guided image synthesis, as several reasons. (1) *Missing global information*. As in Fig. 1(B)-row(b),
vanilla AR models take the top-to-down and left-to-right sequential generation with limited receptive
fields for the initial generating (top left corner), which are incapable of directly modeling global
information. Additionally, the sequential AR models suffer from exposure bias [2], which may

---

*Corresponding author. Dr. Fu is also with Fudan ISTBI—ZJNU Algorithm Centre for Brain-inspired Intelligence, Zhejiang Normal University, Jinhua, China.

35th Conference on Neural Information Processing Systems (NeurIPS 2021).

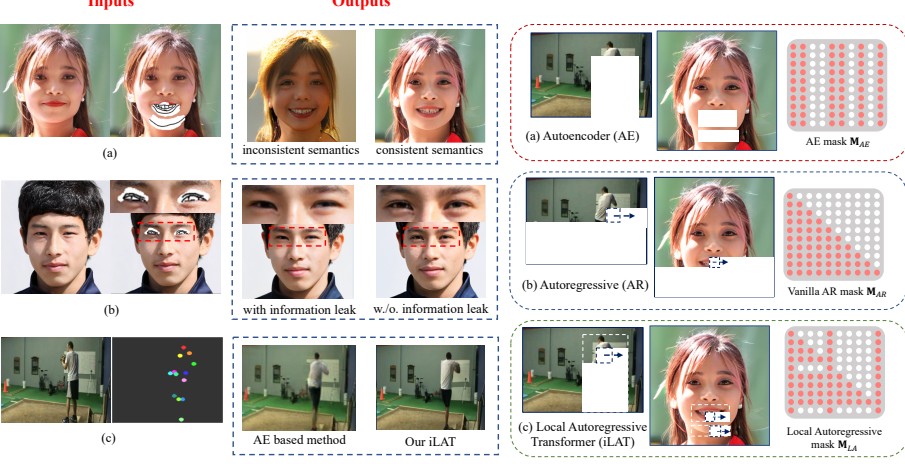

Figure 1: The illustration of (A) influence of missing semantic consistency, the information leak, and blur in AE based method in the local generation, and (B) comparison of AE, AR, and our iLAT for different conditional image generations. Our method is more efficient for locally guided image synthesis by keeping both global semantics and local guidance.

predict future pixels conditioned on the past ones with mistakes, due to the discrepancy between training and testing in AR. This makes small local guidance unpredictable changes to the whole image, resulting in inconsistent semantics as in Fig. 1(A)-row(a). (2) *Slow inference speed*. The AR models have to sequentially predict the pixels in the testing with notoriously slow inference speed, especially for high-resolution image generation. Although the parallel training techniques are used in pixelCNN [32] and transformer [14], the conditional probability sampling fails to work in parallel during the inference phase. (3) *Information leakage of local guidance*. As shown in Fig. 1(B)-row(c), the local guidance should be implemented with specific masks to ensure the validity of the local AR learning. During the sequential training process, pixels from masked regions may be exposed to AR models by convolutions with large kernel sizes or inappropriate attention masks in the transformer. We call it information leakage [44, 16] of local guidance, which makes models overfit the masked regions, and miss detailed local guidance as in Fig. 1(A)-row(b).

To this end, we propose a novel image Local Autoregressive Transformer (iLAT) for the task of locally guided image synthesis. Our key idea lies in learning the local discrete representations effectively. Particularly, we tailor the receptive fields of AR models to local guidance, achieving semantically consistent and visually realistic generation results. Furthermore, a local autoregressive (LA) transformer with the novel LA attention mask and convolution mechanism is proposed to enable successful local generation of images with efficient inference time, without information leakage.

Formally, we propose the iLAT model with several novel components. (1) We complementarily incorporate receptive fields of both AR and AE to fit LA generation with a novel attention mask as shown in Fig. 1(B)-row(c). In detail, local discrete representation is proposed to represent those masked regions, while the unmasked areas are encoded with continuous image features. Thus, we achieve favorable results with both consistent global semantics and realistic local generations. (2) Our iLAT dramatically reduces the inference time for local generation, since only masked regions will be generated autoregressively. (3) A simple but effective two-stream convolution and a local causal attention mask mechanism are proposed for discrete image encoder and transformer respectively, with which information leakage is prevented without detriment to the performance.

We make several contributions in this work. (1) A novel local discrete representation learning is proposed to efficiently help to learn our iLAT for the local generation. (2) We propose an image local autoregressive transformer for local image synthesis, which enjoys both semantically consistent and realistic generative results. (3) Our iLAT only generates necessary regions autoregressively, which is much faster than vanilla AR methods during the inference. (4) We propose a two-stream convolution and a LA attention mask to prevent both convolutions and transformer from information leakage, thus improving the quality of generated images. Empirically, we introduce several locally guidance tasks, including pose-guided image generation and face editing tasks; and extensive experiments are conducted on the corresponding dataset to validate the efficacy of our model.

## 2 Related Work

**Conditional Image Synthesis.** Some conditional generation models are designed to globally generate images with pre-defined styles based on user-provided references, such as poses and face sketches. These previous synthesis efforts are made on Variational auto-encoder (VAE) [10, 13], AR Model [48, 39], and AE with adversarial training [51, 49, 53]. Critically, it is very non-trivial for all these methods to generate images of locally guided image synthesis of the non-iconic foreground. Some tentative attempts have been conducted in pose-guided synthesis [42, 47] with person existing in non-ironic views. On the other hand, face editing methods are mostly based on adversarial AE based inpainting [30, 51, 19] and GAN inversion based methods [53, 40, 1]. Rather than synthesizing the whole image, our iLAT generates the local regions of images autoregressively, which not only improves the stability with the well-optimized joint conditional distribution for large masked regions but also maintains the global semantic information.

**Autoregressive Generation.** The deep AR models have achieved great success recently in the community [35, 15, 32, 9, 38]. Some well known works include PixelRNN [43], Conditional PixelCNN [32], Gated PixelCNN [32], and WaveNet [31]. Recently, transformer based AR models [38, 5, 12] have achieved excellent results in many machine learning tasks. Unfortunately, the common troubles of these AR models are the expensive inference time and potential exposure bias, as AR models sequentially predict future values from the given past values. The inconsistent receptive fields for training and testing will lead to accumulated errors and unreasonable generated results [2]. Our iLAT is thus designed to address these limitations.

**Visual Transformer.** The transformer takes advantage of the self-attention module [44], and shows impressive expressive power in many Natural Language Processing (NLP) [38, 11, 5] and vision tasks [12, 7, 25]. With costly time and space complexity of $O(n^2)$ in transformers, Parmar *et al.* [34] utilize local self-attention to achieve AR generated results. Chen *et al.* [8] and Kumar *et al.* [22] autoregressively generate pixels with simplified discrete color palettes to save computations. But limited by the computing power, they still generate low-resolution images. To address this, some works have exploited the discrete representation learning, *e.g.* dVAE [39] and VQGAN [14]. It not only reduces the sequence length of image tokens but also shares perceptually rich latent features in image synthesis as the word embedding in NLP. However, recovering images from the discrete codebook still causes blur and artifacts in complex scenes. Besides, vanilla convolutions of the discrete encoder may leak information among different discrete codebook tokens. Moreover, local conditional generation based on VQGAN [14] suffers from poor semantical consistency compared with other unchanged image regions. To end these, the iLAT proposes the novel discrete representation to improve the model capability of local image synthesis.

## 3 Approach

Given the conditional image $\mathbf{I}_c$ and target image $\mathbf{I}_t$, our image Local Autoregressive Transformer (iLAT) aims at producing the output image $\mathbf{I}_o$ of semantically consistent and visually realistic. The key foreground objects (*e.g.*, the skeleton of body), or the regions of interest (*e.g.*, sketches of facial regions) extracted from $\mathbf{I}_c$, are applied to guide the synthesis of output image $\mathbf{I}_o$. Essentially, the background and the other non-key foreground image regions of $\mathbf{I}_o$ should be visually similar to $\mathbf{I}_t$. As shown in Fig. 2, our iLAT includes the branches of Two-Stream convolutions based Vector Quantized GAN (TS-VQGAN) for the discrete representation learning and a transformer for the AR generation with Local Autoregressive (LA) attention mask. Particularly, our iLAT firstly encodes $\mathbf{I}_c$ and $\mathbf{I}_t$, into codebook vectors $z_{q,c}$ and $z_{q,t}$ by TS-VQGAN (Sec. 3.1) without local information leakage. Then the index of the masked vectors $\hat{z}_{q,t}$ will be predicted by the transformer autoregressively with LA attention mask (Sec. 3.2). During the test phase, the decoder of TS-VQGAN takes the combination of $\hat{z}_{q,t}$ in masked regions and $\hat{z}$ in unmasked regions to achieve the final result.

### 3.1 Local Discrete Representation Learning

We propose a novel local discrete representation learning in this section. Since it is inefficient to learn the generative transformer model through pixels directly, inspired by VQGAN [14], we incorporate the VQVAE mechanism [33] into the proposed iLAT for the discrete representation learning. The VQVAE is consisted of an encoder $E$, a decoder $D$, and a learnable discrete codebook $\mathbb{Z} = \{z_k\}_{k=1}^{K}$,

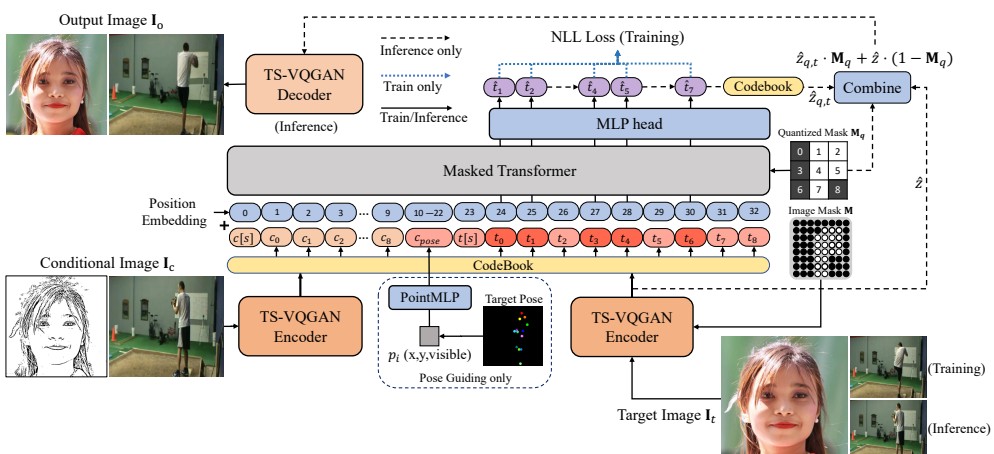

Figure 2: The overview of iLAT with $3 \times 3$ of latent quantization. The target image $\mathbf{I}_t$ is encoded by TS-VQGAN with a binary mask $\mathbf{M}$, while conditional image $\mathbf{I}_c$ is directly encoded without mask. As explained in Sec.3.2, the guided information of poses and sketches are utilized in our two tasks. For pose guidance, target pose coordinates are encoded by a PointMLP into sequential features, which are concatenated with conditional tokens $\{c[s], c_0, ..., c_8, c_{pose}\}$. The TS-VQGAN decoder takes the combination of $\hat{z}_{q,t}$ converted from generated target tokens $\{\hat{t}_1, \hat{t}_2, ..., \hat{t}_7\}$ in masked regions and directly encoded features $\hat{z}$ in unmasked regions to get the final result.

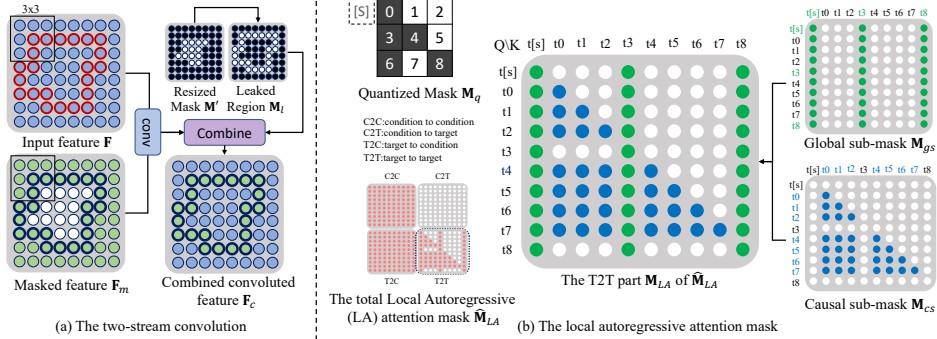

(a) The two-stream convolution          (b) The local autoregressive attention mask

Figure 3: Illustration of (a) the two-stream convolution in the TS-VQGAN encoder and (b) the LA attention mask $\hat{\mathbf{M}}_{LA}$. In (a), after the feature $\mathbf{F}$ is convoluted, the $3 \times 3$ convolution kernel spreads the features from the masked regions to unmasked regions. Patches with leaked information are circled with red. For (b), a $3 \times 3$ quantized mask $\mathbf{M}_q$ is assumed as the input mask. $\mathbf{M}_{LA}$ can be divided into the global sub-mask $\mathbf{M}_{gs}$ and the causal sub-mask $\mathbf{M}_{cs}$. Then, $t[s], t_3, t_8$ can be categorized into global pixel tokens, while others are casual tokens: $t_1, t_2, t_5, t_7$ are masked tokens $t_m, (\mathbf{M}_{q,m} = 1)$, and $t_0, t_1, t_4, t_6$ are tokens $t_{m-1}$ that need to predict masked ones. The global tokens can be attended with all tokens. And causal tokens attend to the targets with a lower triangular matrix. All colored tokens are valued 1 and white tokens are 0 in $\mathbf{M}_{LA}$.

where $K$ means the total number of codebook vectors. Given an image $\mathbf{I} \in \mathbb{R}^{H \times W \times 3}$, $E$ encodes the image into latent features $\hat{z} = E(\mathbf{I}) \in \mathbb{R}^{h \times w \times c_e}$, where $c_e$ indicates the channel of the encoder outputs. Then, the spatially unfolded $\hat{z}_{h'w'} \in \mathbb{R}^{c_e}, (h' \in h, w' \in w)$ are replaced with the closest codebook vectors as

$$z_{h'w'}^{(q)} = \arg \min_{z_k \in \mathbb{Z}} ||\hat{z}_{h'w'} - z_k|| \in \mathbb{R}^{c_q},$$

$$z_q = \text{fold}(z_{h'w'}^{(q)}, h' \in h, w' \in w) \in \mathbb{R}^{h \times w \times c_q}, \tag{1}$$

where $c_q$ indicates the codebook channels. However, VQVAE will suffer from obscure information leakage for the local image synthesis, if the receptive field (kernel size) of vanilla convolution is larger than $1 \times 1$ as shown in Fig. 3(a). Intuitively, each $3 \times 3$ convolution layer spreads the masked features to the outside border of the mask. Furthermore, multi-convolutional based $E$ accumulates the information leakage, which makes the model learn the local generating with unreasonable confidence, leading to model overfitting (see Sec. 4.3).

To this end, we present two-stream convolutions in TS-VQGAN as shown in Fig. 3(a). Since the masked information is only leaked to a circle around the mask with each $3 \times 3$ convolution layer, we can just replace the corrupt features for each layer with masked ones. Thus, the influence of information leakage will be eliminated without hurting the integrity of both masked and unmasked features. Specifically, for the given image mask $\mathbf{M} \in \mathbb{R}^{H \times W}$ that 1 means masked regions, and 0 means unmasked regions, it should be resized into $\mathbf{M}'$ with max-pooling to fit the feature size. The two-stream convolution converts the input feature $\mathbf{F}$ into the masked feature $\mathbf{F}_m = \mathbf{F} \odot (1 - \mathbf{M}')$, where $\odot$ is element-wise multiplication. Then, both $\mathbf{F}$ and $\mathbf{F}_m$ are convoluted with shared weights and combined according to the leaked regions $\mathbf{M}_l$, which can be obtained from the difference of convoluted mask as

$$\mathbf{M}_l = \mathrm{clip}(\mathrm{conv}_\mathbf{1}(\mathbf{M}'), 0, 1) - \mathbf{M}', \quad \mathbf{M}_l[\mathbf{M}_l > 0] = 1, \tag{2}$$

where $\mathrm{conv}_\mathbf{1}$ implemented with an all-one $3 \times 3$ kernel. Therefore, the output of two-stream convolution can be written as

$$\mathbf{F}_c = \mathrm{conv}(\mathbf{F}) \odot (1 - \mathbf{M}_l) + \mathrm{conv}(\mathbf{F}_m) \odot \mathbf{M}_l. \tag{3}$$

So the leaked regions are replaced with features that only depend on unmasked regions. Besides, masked features can be further leveraged for AR learning without any limitations. Compared with VQVAE, we replace all vanilla convolutions with two-stream convolutions in the encoder of TS-VQGAN. Note that the decoder $D$ is unnecessary to prevent information leakage at all. Since the decoding process is implemented after the AR generating of the transformer as shown in Fig. 2.

For the VQVAE, the decoder $D$ decodes the codebook vectors $z_q$ got from Eq.(1), and reconstructs the output image as $\mathbf{I}_o = D(z_q)$. Although VQGAN [14] can generate more reliable textures with adversarial training, handling complex real-world backgrounds and precise face details is still tough to the existed discrete learning methods. In TS-VQGAN, we further finetune the model with local quantized learning, which can be written as

$$\mathbf{I}_o = D(z_q \odot \mathbf{M}_q + \hat{z} \odot (1 - \mathbf{M}_q)), \tag{4}$$

where $\mathbf{M}_q \in \mathbb{R}^{h \times w}$ is the resized mask for quantized vectors, and $\hat{z}$ is the output of the encoder. In Eq.(4), unmasked features are directly encoded from the encoder, while masked features are replaced with the codebook vectors, which works between AE and VQVAE. This simple trick effectively maintains the fidelity of the unmasked regions and reduces the number of quantized vectors that have to be generated autoregressively, which also leads to a more efficient local AR inference. Note that the back-propagation of Eq.( 4) is implemented with the straight-through gradient estimator [4].

## 3.2 Local Autoregressive Transformer Learning

From the discrete representation learning in Sec. 3.1, we can get the discrete codebook vectors $z_{q,c}, z_{q,t} \in \mathbb{R}^{h \times w \times c_q}$ for conditional images and target images respectively. Then the conditional and target image tokens $\{c_i, t_j\}_{i,j=1}^{hw} \in \{0, 1, ..., K - 1\}$ can be converted from the index-based representation of $z_{q,c}, z_{q,t}$ in the codebook with length $hw$, where $K$ indicates the all number of codebook vectors. For the resized target mask $\mathbf{M}_q \in \mathbb{R}^{h \times w}$, the second stage needs to learn the AR likelihood for the masked target tokens $\{t_m\}$ where $\mathbf{M}_{q,m} = 1$ with conditional tokens $\{c_i\}_{i=1}^{hw}$ and other unmasked target tokens $\{t_u\}$ where $\mathbf{M}_{q,u} = 0$ as

$$p(t_m|c, t_u) = \prod_j p(t_{(m,j)}|c, t_u, t_{(m,<j)}). \tag{5}$$

Benefits from Eq. (4), iLAT only needs to generate masked target tokens $\{t_m\}$ rather than all. Then, the negative log likelihood (NLL) loss can be optimized as

$$\mathcal{L}_{NLL} = -\mathbb{E}_{t_m \sim p(t_m|c,t_u)} \log p(t_m|c, t_u). \tag{6}$$

We use a decoder-only transformer to handle the AR likelihood. As shown in Fig. 2, two special tokens $c[s], t[s]$ are concatenated to $\{c_i\}$ and $\{t_j\}$ as start tokens respectively. Then, the trainable position embedding [11] is added to the token embedding to introduce the position information to the self-attention modules. According to the self-attention mechanism in the transformer, the attention mask is the key factor to achieve parallel training without information leakage. As shown in Fig. 3(b), we propose a novel LA attention mask $\hat{\mathbf{M}}_{LA}$ with four sub-masks, which indicate receptive fields

of condition to condition (C2C), condition to target (C2T), target to condition (T2C), and target to target (T2T) respectively. All conditional tokens $\{c_i\}_{i=1}^{hw}$ can be attended by themselves and targets. So C2C and T2C should be all-one matrices. We think that the conditional tokens are unnecessary to attend targets in advance, so C2T is set as the all-zero matrix. Therefore, the LA attention mask $\hat{\mathbf{M}}_{LA}$ can be written as

$$\hat{\mathbf{M}}_{LA} = \begin{bmatrix} \mathbf{1}, & \mathbf{0} \\ \mathbf{1}, & \mathbf{M}_{LA} \end{bmatrix}, \tag{7}$$

where $\mathbf{M}_{LA}$ indicates the T2T LA mask. To leverage the AR generation and maintain the global information simultaneously, the target tokens are divided into two groups called the global group and the causal group. Furthermore, the causal group includes masked targets $\{t_m\}(\mathbf{M}_{q,m} = 1)$ and $\{t_{m-1}\}$ that need to predict them, because the labels need to be shifted to the right with one position for the AR learning. Besides, other tokens are classified into the global group. Then, the global attention sub-mask $\mathbf{M}_{gs}$ can be attended to all tokens to share global information and handle the semantic consistency. On the other hand, the causal attention sub-mask $\mathbf{M}_{cs}$ constitutes the local AR generation. Note that $\mathbf{M}_{gs}$ can not attend any masked tokens to avoid information leakage. The T2T LA mask can be got with $\mathbf{M}_{LA} = \mathbf{M}_{gs} + \mathbf{M}_{cs}$[2]. A more intuitive example is shown in Fig. 3(b). Therefore, for the given feature $h$, the self-attention in iLAT can be written as

$$\text{SelfAttention}(h) = \text{softmax}(\frac{QK^T}{\sqrt{d}} - (1 - \hat{\mathbf{M}}_{LA}) \cdot \infty)V, \tag{8}$$

where $Q, K, V$ are $h$ encoded with different weights of $d$ channels. We make all masked elements to $-\infty$ before the softmax. During the inference, all generated target tokens $\{\hat{t}_m\}$ are converted back to codebook vectors $\hat{z}_{q,t}$. Then, they are further combined with encoded unmasked features $\hat{z}$, and decoded with Eq.(4) as shown in Fig. 2.

To highlight the difference of our proposed mask $\mathbf{M}_{LA}$, other common attention masks are shown in Fig. 1. The Vanilla AR mask $\mathbf{M}_{AR}$ is widely used in the AR transformer [14, 8], but they fail to maintain the semantic consistency and cause unexpected identities for the face synthesis. AE mask $\mathbf{M}_{AE}$ is utilized in some attention based image inpainting tasks [50, 49]. Although $\mathbf{M}_{AE}$ enjoys good receptive fields, the masked regions are completely corrupted in the AE, which is much more unstable to restructure a large hole. Our method is an in-between strategy with both their superiorities mentioned above.

### 3.3 Implement Details for Different Tasks

**Non-Iconic Posed-Guiding.** The proposed TS-VQGAN is also learned with adversarial training. For the complex non-iconic pose guiding, we finetune the pretrained open-source ImageNet based VQGAN weights with the two-stream convolution strategy. To avoid adding too many sequences with excessive computations, we use the coordinates of 13 target pose landmarks as the supplemental condition to the iLAT. They are encoded with 3 fully connected layers with ReLU. As shown in Fig. 2, both the condition and the target are images, which have different poses in the training phase, and the same pose in the inference phase. Besides, we use the union of conditional and target masks got by dilating the poses with different kernel sizes according to the scenes[3] to the target image.

**Face Editing.** In face editing, we find that the adaptive GAN learning weight $\lambda$ makes the results unstable, so it is replaced with a fixed $\lambda = 0.1$. Besides, the TS-VQGAN is simplified compared to the one used in the pose guiding. Specifically, all attention layers among the encoder and decoder are removed. Then, all Group Normalizations are replaced with Instance Normalization to save memory without a large performance drop. The conditions are composed of the sketch images extracted with the XDoG [46], while the targets are face images. The training masks for the face editing are COCO masks [24] and simulated irregular masks [51], while the test ones are drawn manually.

## 4 Experiments

In this section, we present experimental results on pose-guided generation of Penn Action (PA) [52] and Synthetic DeepFashion (SDF) [26], face editing of CelebA [27] and FFHQ [20] compared with other competitors and variants of iLAT.

---

[2]More about the expansion of LA attention mask are discussed in the supplementary.
[3]Details about the mask generation are illustrated in the supplementary.

Table 1: Quantitative results in PA (left) and SDF (right). ↑ means larger is better while ↓ means lower is better. iLAT* indicates that iLAT trained without two-stream convolutions.

| | PATN | PN-GAN | Posewarp | MR-Net | Taming | iLAT* | iLAT | Taming | iLAT |
|---|---|---|---|---|---|---|---|---|---|
| PSNR↑ | 20.83 | 21.36 | 21.76 | 21.79 | 21.43 | 21.68 | **22.94** | 16.25 | **16.71** |
| SSIM↑ | 0.744 | 0.761 | 0.794 | 0.792 | 0.746 | 0.748 | **0.800** | 0.539 | **0.599** |
| MAE↓ | 0.062 | 0.062 | 0.053 | 0.066 | 0.057 | 0.056 | **0.046** | 0.107 | **0.096** |
| FID↓ | 82.79 | 64.43 | 93.61 | 79.50 | 33.53 | 31.83 | **27.36** | 72.77 | **70.58** |

**Datasets.** For the pose guiding, PA dataset [52], which contains 2,326 video sequences of 15 action classes in non-iconic views is used in this section. Each frame from the video is annotated with 13 body landmarks consisted of 2D locations and visibility. The resolution of PA is resized into $256 \times 256$ during the preprocessing. We randomly gather pairs of the same video sequence in the training phase dynamically and select 1,000 testing pairs in the remaining videos. Besides, the SDF is synthesized with DeepFashion [26] images as foregrounds and Places2 [54] images as backgrounds. Since only a few images of DeepFashion have related exact segmentation masks, we select 4,500/285 pairs from it for training and testing respectively. Each pair of them contains two images of the same person with different poses and randomly chosen backgrounds. The face editing dataset consists of Flickr-Faces-HQ dataset (FFHQ) [20] and CelebA-HQ [27]. FFHQ is a high-quality image dataset with 70,000 human faces. We resize them from $1024 \times 1024$ into $256 \times 256$ and use 68,000 of them for the training. The CelebA is only used for testing in this section for the diversity. Since face editing has no paired ground truth, we randomly select 68 images from the rest of FFHQ and all CelebA, and draw related sketches for them.

**Implementation Details.** Our method is implemented in PyTorch in $256 \times 256$ image size. For the TS-VQGAN training, we use the Adam optimizer [21] with $\beta_1 = 0.5$ and $\beta_2 = 0.9$. For the pose guiding, the TS-VQGAN is finetuned from the ImageNet pretrained VQGAN [14], while it is completely retrained for FFHQ. TS-VQGAN is trained with $150k$ steps without masks at first, and then it is trained with another $150k$ steps with masks in batch size 16. The initial learning rates of pose guiding and face editing are $8e$-5 and $2e$-4 respectively, which are decayed by 0.5 for every $50k$ steps. For the transformer training, we use Adam with $\beta_1 = 0.9$ and $\beta_2 = 0.95$ with initial learning rate $5e$-5 and 0.01 weight decay. Besides, we warmup the learning rate with the first $10k$ steps, then it is linearly decayed to 0 for $300k$ iterations with batch size 16. During the inference, we simply use top-1 sampling for our iLAT.

**Competitors.** The model proposed in [14] is abbreviated as Taming transformer (Taming) in this section. For fair comparisons, VQGAN used in Taming is finetuned for pose guiding, and retrained for face editing with the same steps as TS-VQGAN. For the pose guiding, we compare the proposed iLAT with other state-of-the-art methods retrained in the PA dataset, which include PATN [56], PN-GAN [37], PoseWarp [3], MR-Net [47] and Taming [14]. As the image size of PoseWarp and MR-Net is $128 \times 128$, we resized the outputs for the comparison. For the face editing, we compare the iLAT with inpainting based SC-FEGAN [19] and Taming [14]. We also test the Taming results in our LA attention mask as Taming* (without retraining).

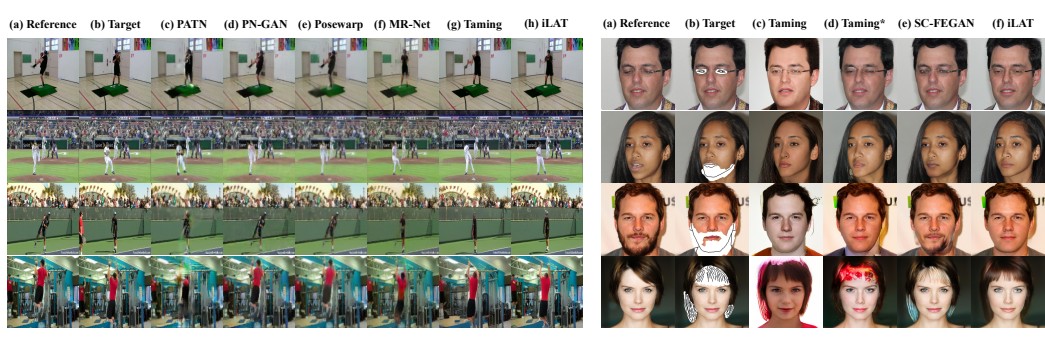

(A) Pose-Guided Generation in PA.  (B) FFHQ (row 1, 2) and CelebA (row 3, 4).

Figure 4: Qualitative results. Targets in (B) are combined with masks and XDoG sketches. Taming* means that the Taming transformer tested with our LA attention mask. Please zoom-in for details.

Table 2: Average inference time (sec/image) in PA, SDF, and FFHQ of the vanilla AR transformer based generation (Taming) and iLAT. We also show the average masked rate of three datasets.

|  | masked rate | Taming | iLAT |
| --- | --- | --- | --- |
| PA | 31.97% | 8.551 | **3.426** |
| SDF | 28.09% | 8.372 | **3.898** |
| FFHQ | 6.64% | 8.183 | **1.180** |

## 4.1 Quantitative Results

**Pose-Guided Comparison.** Quantitative results in PA and SDF datasets of our proposed iLAT and other competitors are presented in Tab. 1. Peak signal-to-noise ratio (PSNR), Structural Similarity (SSIM) [45], Mean Average Error (MAE) and Fréchet Inception Distance (FID) [17] are employed to measure the quality of results. We also add the results of iLAT*, which is implemented without the two-stream convolutions. The results in Tab. 1 clearly show that our proposed method outperforms other methods in all metrics, especially for the FID, which accords with the human perception. The good scores of iLAT indicate that the proposed iLAT can generate more convincing and photo-realistic images on locally guided image synthesis of the non-iconic foreground. For the more challenging SDF dataset, iLAT still achieves better results compared with Taming.

**Inference Time.** We also record the average inference times in PA, SDF, and FFHQ as showed in Tab. 2. Except for the better quality of generated images over Taming as discussed above, our iLAT costs less time for the local synthesis task according to the masking rate of the inputs. Low masking rates can achieve dramatic speedup, *e.g.*, face editing.

## 4.2 Qualitative Results

**Non-Iconic Pose Guiding.** Fig. 4(A) shows qualitative results in the non-iconic pose-guided image synthesis task. Compared to other competitors, it is apparent that our method can generate more reasonable target images both in human bodies and backgrounds, while images generated by other methods suffer from either wrong poses or deformed backgrounds. Particularly, PATN collapses in most cases. PN-GAN and PoseWarp only copy the reference images as the target ones, which fails to be guided by the given poses due to the challenging PA dataset. Moreover, MR-Net and Taming* can indeed generate poses that are similar to the target ones, but the background details of reference images are not transferred properly. Especially for the results in column (g), Taming fails to synthesize complicated backgrounds, such as noisy audiences in rows 2 and 3 and the gym with various fitness equipment in row 4. Compared to others, our proposed iLAT can capture the structure of human bodies given the target poses as well as retaining the vivid backgrounds, which demonstrate the efficacy of our model in synthesizing high-quality images in the non-iconic pose guiding. Besides, for the pose guiding with synthetic backgrounds of SDF, iLAT can still get more reasonable and stable backgrounds and foregrounds compared with Taming as in Fig. 5(C).

**Face Editing.** Since there are no ground truth face editing targets, we only compared the qualitative results as shown in Fig. 4(B) of FFHQ and CelebA. Note that the Taming results in column (c) fail to preserve the identity information in both FFHQ and CelebA compared with the reference. For example, in rows 1, 2 and 3, the skin tones of Taming results are different from the original ones. And in row 4, Taming generates absolutely another person with contrasting ages, which indicates that vanilla AR is unsuited to the local face editing. When Taming is tested with our LA attention mask, column (d) shows that Taming* can retain the identities of persons. However, rows 1 and 2 demonstrate that Taming* fails to properly generate the target faces according to guided sketches, while in rows 3 and 4 some generations have obvious artifacts without consistency. Besides, inpainting-based SC-FEGAN achieves unstable results in rows 3 and 4. SC-FEGAN also strongly depends on the quality of input sketches, while unprofessional sketches lead to unnatural results as shown in row 1. Besides, detailed textures of AE-based SC-FEGAN are unsatisfactory too. Compared with these methods, our iLAT can always generate correct and vivid human faces with identities retained. Furthermore, benefits from the discrete representation, iLAT enjoys robustness to the guided information.

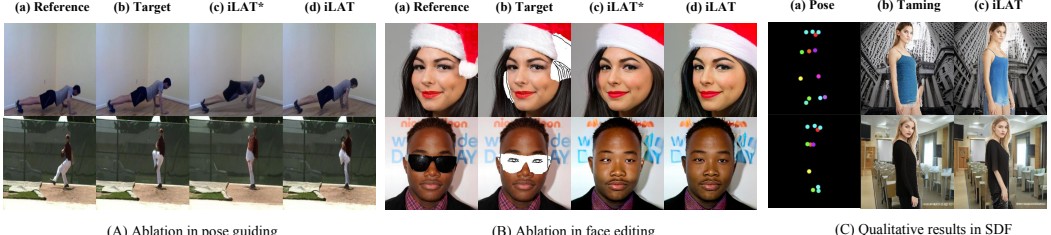

| (a) Reference | (b) Target | (c) iLAT* | (d) iLAT | (a) Reference | (b) Target | (c) iLAT* | (d) iLAT | (a) Pose | (b) Taming | (c) iLAT |

| (A) Ablation in pose guiding | (B) Ablation in face editing | (C) Qualitative results in SDF |

Figure 5: Ablation study for two-stream convolutions (A, B) and qualitative results in SDF (C). iLAT* means iLAT without two-stream convolutions. Please zoom-in for details.

## 4.3 Further Discussions

**Ablation Study.** The effectiveness of our proposed two-stream convolution is discussed in the ablation study. As we can find in Fig. 5, the woman face in row 1, (c) generated by iLAT* has residual face parts that conflict with the guided sketch. Moreover, in row 2, iLAT* without two-stream convolutions leaks information from sunglasses that lacks correct semantic features and leads to the inconsistent color of the generated face. For the pose-guided instance shown in the second row, it is apparent that the man generated in column (c) has blurry leg positions. However, in column (d) the complete iLAT can always generate authentic and accurate images, validating the efficacy of our designed two-stream convolutions.

**Sequential Generation.** Our iLAT can also be extended to guide the video generation properly. We give a qualitative example in this section. As shown in Fig. 6, given a sequence of poses and one reference image, iLAT can forecast a plausible motion of the person. And the results are robust for most kinds of activities in non-ironic views.

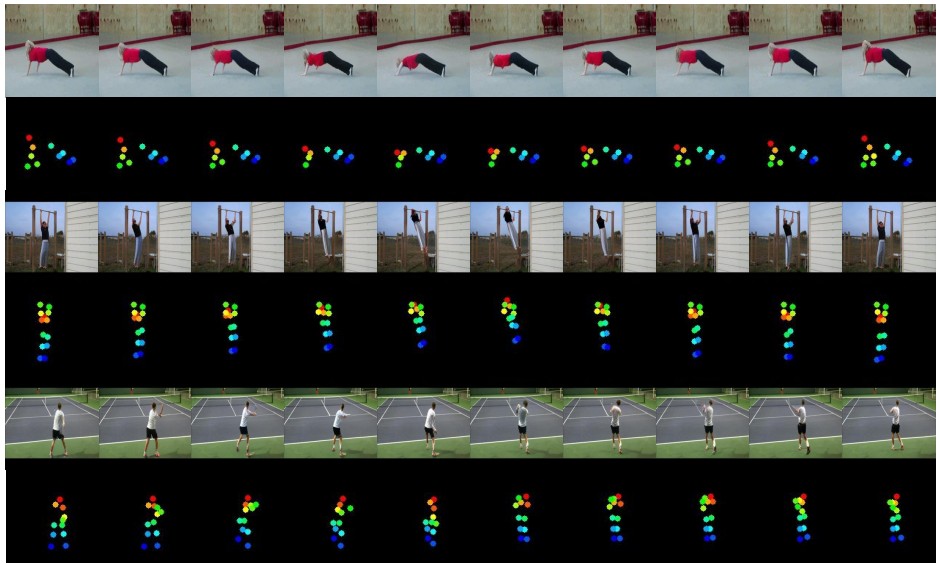

Figure 6: Sequential generated results by iLAT in different ironic views, where odd rows mean the generated images, while even rows indicate the guided poses.

## 5 Conclusion

This paper proposes a transformer based AR model called iLAT to solve local image generation tasks. This method leverages a novel LA attention mask to enlarge the receptive fields of AR, which achieves not only semantically consistent but also realistic generative results and accelerates the inference speed of AR. Besides, a TS-VQGAN is proposed to learn a discrete representation learning without information leakages. Such a model can get superior performance in detail editing. Extensive experiments validate the efficacy of our iLAT for local image generation.

## Social Impacts

This paper exploited the image editing with transformers. Since face editing may causes some privacy issues, we sincerely remind users to pay attention for it. Our method only focuses on technical aspects. The images used in this paper are all open sourced.

## Acknowledgements

This work was supported in part by NSFC Project (62076067, 62176061), Science and Technology Commission of Shanghai Municipality Projects (19511120700, 2021SHZDZX0103).

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
