# Supplementary Materials of The Image Local Autoregressive Transformer

**Chenjie Cao, Yuxin Hong, Xiang Li, Chengrong Wang, Chengming Xu, Yanwei Fu, Xiangyang Xue**
School of Data Science
Fudan University
{20110980001,yanweifu}@fudan.edu.cn

## 1 Network Architectures

**TS-VQGAN.** The architecture of TS-VQGAN used in pose guiding (Penn [11] and synthesis Deep-Fashion [7]) is follow the VQGAN [4] with its ImageNet pretrained weights. The TS-VQGAN for face datasets (FFHQ [6]) is simplified as in Tab. 1. The channels of codebook are 256. For the pose guiding and face editing, we set the total codebook number as 16384 and 2048 respectively.

**Transformer.** We use the transformer with a 'base' scale [3] of channels 768, attention head numbers 12, transformer layers 12, which is trained with dropout rate 0.1.

## 2 Supplementary Experiment Details

Table 1: TS-VQGAN for face datasets, where Two-stream (TS) convolutions are only used in the encoder.

| Encoder | Decoder |
|---|---|
| TS-Conv2d ($256 \times 256 \times 64$) | Conv2d ($16 \times 16 \times 512$) |
| ResidualBlock$\times$2+DownBlock ($128 \times 128 \times 64$) | ResidualBlock$\times$4 ($16 \times 16 \times 512$) |
| ResidualBlock$\times$2+DownBlock ($64 \times 64 \times 128$) | ResidualBlock$\times$2+UpBlock ($32 \times 32 \times 384$) |
| ResidualBlock$\times$2+DownBlock ($32 \times 32 \times 256$) | ResidualBlock$\times$2+UpBlock ($64 \times 64 \times 256$) |
| ResidualBlock$\times$2+DownBlock ($16 \times 16 \times 384$) | ResidualBlock$\times$2+UpBlock ($128 \times 128 \times 128$) |
| ResidualBlock$\times$4 ($16 \times 16 \times 512$) | ResidualBlock$\times$2+UpBlock ($256 \times 256 \times 64$) |
| InstanceNorm+SWISH ($16 \times 16 \times 512$) | InstanceNorm+SWISH ($256 \times 256 \times 64$) |
| TS-Conv2d ($16 \times 16 \times 256$) | Conv2d+Tanh ($256 \times 256 \times 3$) |

| Encoder ResidualBlock | Decoder ResidualBlock | DownBlock | UpBlock |
|---|---|---|---|
| InstanceNorm+SWISH | InstanceNorm+SWISH | TS-Conv2d (stride=2) | Nearest Upsample |
| TS-Conv2d | Conv2d | – | Conv2d |
| InstanceNorm+SWISH | InstanceNorm+SWISH | – | – |
| TS-Conv2d | Conv2d | – | – |

### 2.1 Masks for the Penn Action Dataset

Since the Penn Action (PA) [11] dataset includes various scenes videos, many handheld objects influence the generated results, *e.g.*, baseball bats, golf clubs, and barbells, etc. We get the related masks with various dilated rates according to the image scenes as shown in Fig. 1.

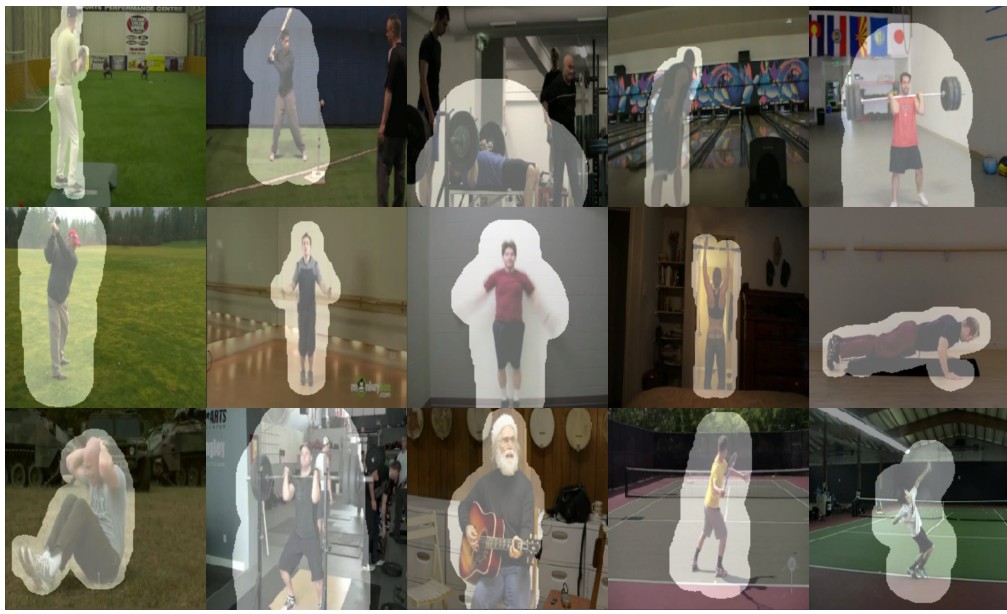

Figure 1: The illustration of masks for PA in different scenes. The masking dilated rates from left to right, up to bottom are, 'baseball_pitch':15, 'baseball_swing':60, 'bench_press':120, 'bowl':20, 'clean_and_jerk':120, 'golf_swing':70, 'jump_rope':30, 'jumping_jacks':60, 'pullup':30, 'pushup':30, 'situp':25, 'squat':120, 'strum_guitar':25, 'tennis_forehand':60, 'tennis_serve':60.

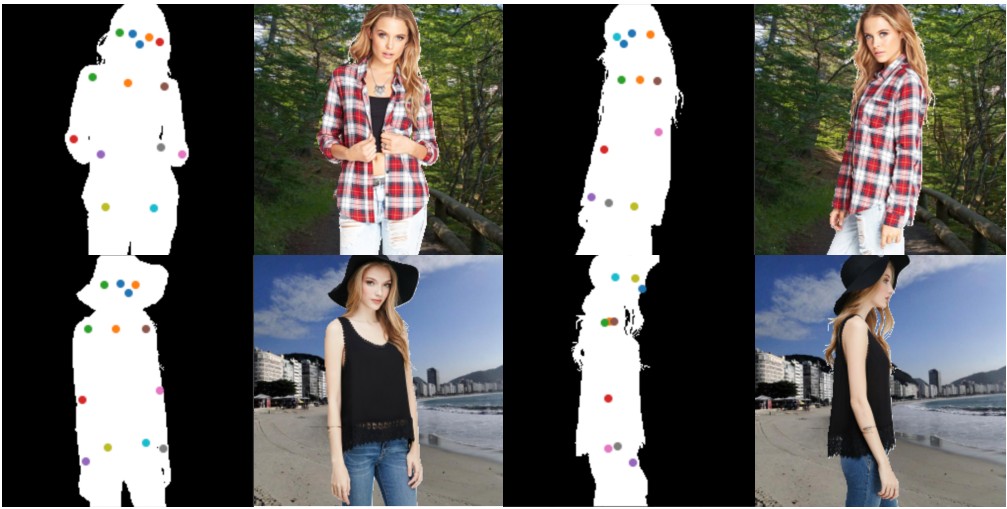

Figure 2: The illustration of the SDF dataset. Columns 1 and 3 are masks and pose landmarks (18 landmarks with -1 indicating invisible points), while columns 2 and 4 are related synthetic pictures.

## 2.2   Synthesis for the DeepFashion and Places2

The synthesis DeepFashion [7] (SDF) is composed of DeepFashion and Places2 [12] as foregrounds and backgrounds respectively. We show the combined results in Fig. 2.

Table 2: Human evaluation in face editing.

|  | Taming | Taming* | SC-FEGAN | iLAT |
|---|---|---|---|---|
| Average Scores | 0.33 | 3.33 | 17.33 | **38.33** |

## 3 More Experiment Results

### 3.1 Human Evaluation

For more comprehensive comparisons, 68 images selected from FFHQ and CelebA are compared by 3 uncorrelated volunteers. Particularly, volunteers need to choose the best one from randomly shuffled 4 compared methods, which include Taming [4], Taming* (Taming with LA mask), SC-FEGAN [5], and our iLAT. Outputs of all these methods are combined with the references. Then, they give one score to the best approach. If all methods work roughly the same for a certain sample, it will be ignored for the final decision. The average scores are shown in Tab. 2.

### 3.2 Combining Outputs and References

Although our proposed iLAT can maintain the global semantics in most cases, it still suffers from few unexpected changes in some details, *e.g.*, facial features (Fig. 3 rows 1 and 3 (c)), words in backgrounds (Fig. 3 row 2 (c)), and hats (Fig. 3 row 4 (c)) as shown in Fig. 3. This problem is similar to the one in most GAN inverse based methods [1, 9]. To this end, these methods combine their outputs with original images, and achieve the final results. We also show the combination of our method in face dataset in Fig. 3. We simply resize the input masks to $128 \times 128$, and resize them back to $256 \times 256$ for the smoothness. Note that combinations of Taming and references have obvious boundaries caused by the inconsistent semantics (Fig. 3 (d)), while our iLAT outputs can be seamlessly combined with the references, and achieve much better performance.

### 3.3 More Qualitative Results

We show more related qualitative experiment results in face dataset (Fig. 4), PA (Fig. 5), SDF (Fig. 6), and sequential generations of pose guiding (Fig. 7).

### 3.4 Alternative Version of the Attention Mask

As discussed in the main paper, the proposed local autoregressive (LA) attention mask $\mathbf{M}_{LA}$ enjoys both global semantic information and local AR generation.

In contrast, we also consider another possible version for attention mask as illustrated in Fig. 8: For the tokens that are not needed to predict any other tokens, such as $\{t[s], t_2, t_3, t_5, t_7, t_8\}$, their respective fields can be further shifted to right until the masked tokens, as the red points shown in Fig. 8. However, technically it is very hard to implement this version processed in parallel; and thus it is impossible to efficiently utilize this version in our iLAT. We will make it as the future work of extending this alternative of attention mask.

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

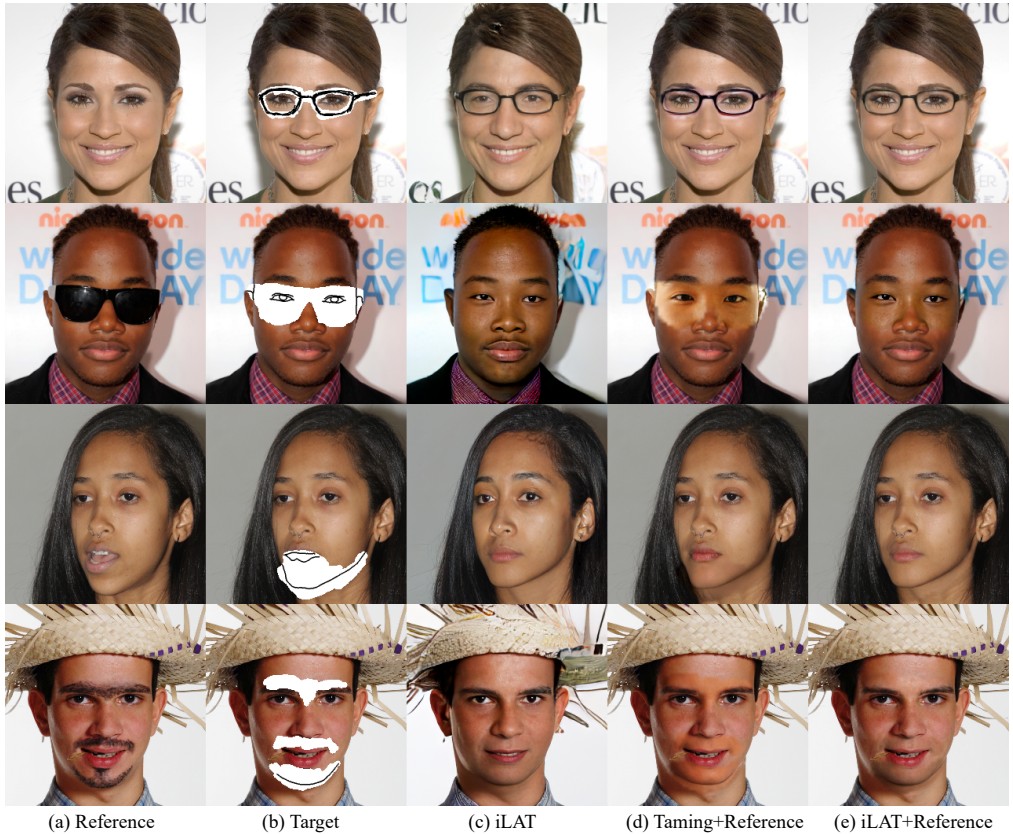

| (a) Reference | (b) Target | (c) iLAT | (d) Taming+Reference | (e) iLAT+Reference |

Figure 3: Comparisons of face dataset, where images in (d) are combinations of Taming and references, and images in (e) are combinations of iLAT and references. Please zoom-in for details.

[4] Patrick Esser, Robin Rombach, and Björn Ommer. Taming transformers for high-resolution image synthesis. *arXiv preprint arXiv:2012.09841*, 2020.

[5] Youngjoo Jo and Jongyoul Park. Sc-fegan: face editing generative adversarial network with user's sketch and color. In *Proceedings of the IEEE/CVF International Conference on Computer Vision*, pages 1745–1753, 2019.

[6] Tero Karras, Samuli Laine, and Timo Aila. A style-based generator architecture for generative adversarial networks. In *Proceedings of the IEEE/CVF Conference on Computer Vision and Pattern Recognition*, pages 4401–4410, 2019.

[7] Ziwei Liu, Ping Luo, Shi Qiu, Xiaogang Wang, and Xiaoou Tang. Deepfashion: Powering robust clothes recognition and retrieval with rich annotations. In *Proceedings of IEEE Conference on Computer Vision and Pattern Recognition (CVPR)*, June 2016.

[8] Xuelin Qian, Yanwei Fu, Tao Xiang, Wenxuan Wang, Jie Qiu, Yang Wu, Yu-Gang Jiang, and Xiangyang Xue. Pose-normalized image generation for person re-identification. In *Proceedings of the European conference on computer vision (ECCV)*, pages 650–667, 2018.

[9] Elad Richardson, Yuval Alaluf, Or Patashnik, Yotam Nitzan, Yaniv Azar, Stav Shapiro, and Daniel Cohen-Or. Encoding in style: a stylegan encoder for image-to-image translation. *arXiv preprint arXiv:2008.00951*, 2020.

[10] Chengming Xu, Yanwei Fu, Chao Wen, Ye Pan, Yu-Gang Jiang, and Xiangyang Xue. Pose-guided person image synthesis in the non-iconic views. *IEEE Transactions on Image Processing*, 29:9060–9072, 2020.

| (a) Reference | (b) Target | (c) Taming | (d) Taming* | (e) SC-FEGAN | (f) iLAT | (g) iLAT+Ref |
|---|---|---|---|---|---|---|

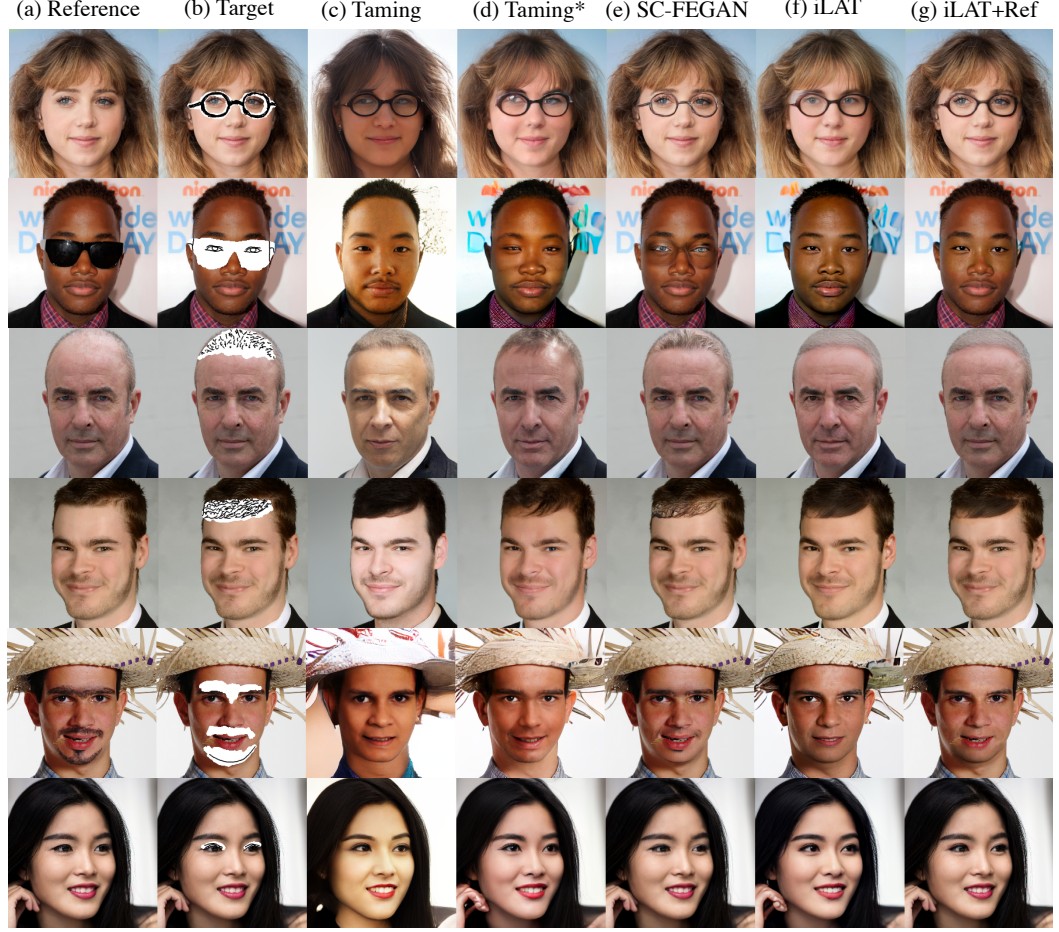

Figure 4: More results from CelebA-HQ and FFHQ. From left to right, references, targets with sketches, Taming [4], Taming with LA mask (Taming*), SC-FEGAN [5], our iLAT, iLAT combined with references.

[11] Weiyu Zhang, Menglong Zhu, and Konstantinos G Derpanis. From actemes to action: A strongly-supervised representation for detailed action understanding. In *Proceedings of the IEEE International Conference on Computer Vision*, pages 2248–2255, 2013.

[12] Bolei Zhou, Agata Lapedriza, Aditya Khosla, Aude Oliva, and Antonio Torralba. Places: A 10 million image database for scene recognition. *IEEE Transactions on Pattern Analysis and Machine Intelligence*, 2017.

[13] Zhen Zhu, Tengteng Huang, Baoguang Shi, Miao Yu, Bofei Wang, and Xiang Bai. Progressive pose attention transfer for person image generation. In *Proceedings of the IEEE/CVF Conference on Computer Vision and Pattern Recognition*, pages 2347–2356, 2019.

| (a) Reference | (b) Target | (c) PATN | (d) PN-GAN | (e) Posewarp | (f) MR-Net | (g) Taming | (h) iLAT |
| --- | --- | --- | --- | --- | --- | --- | --- |

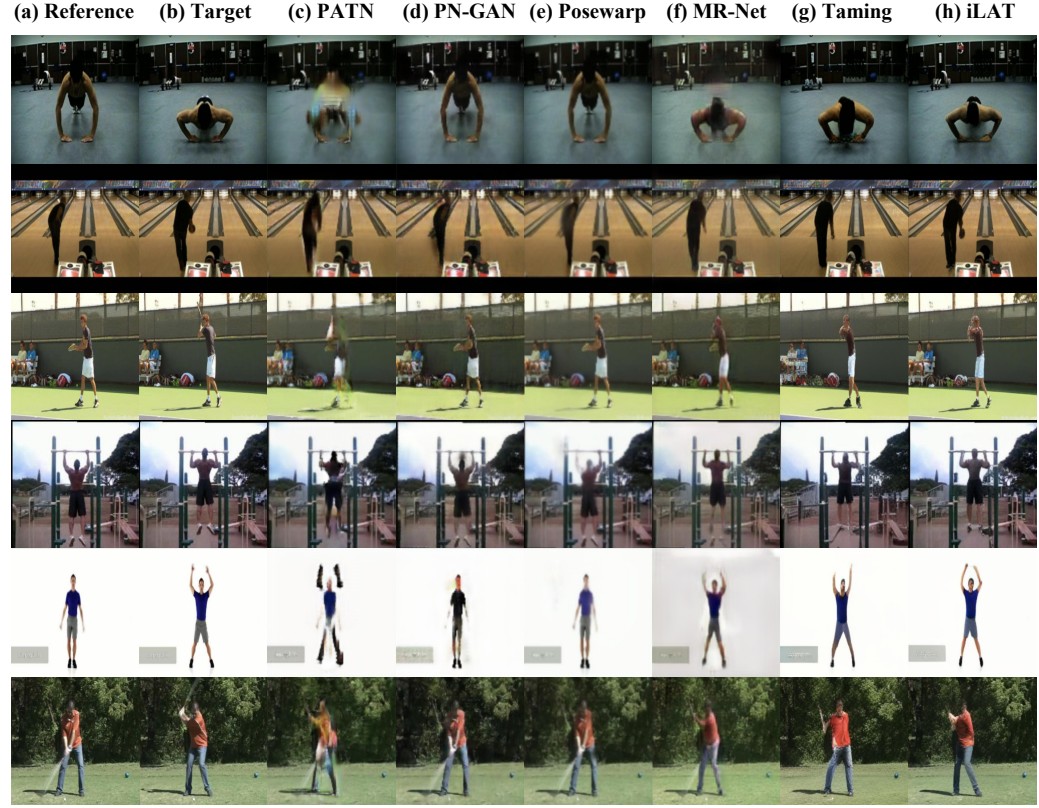

Figure 5: More results from PA. From left to right, references, targets, PATN [13], PN-GAN [8], PoseWarp [2], MR-Net [10], Taming [4], our iLAT.

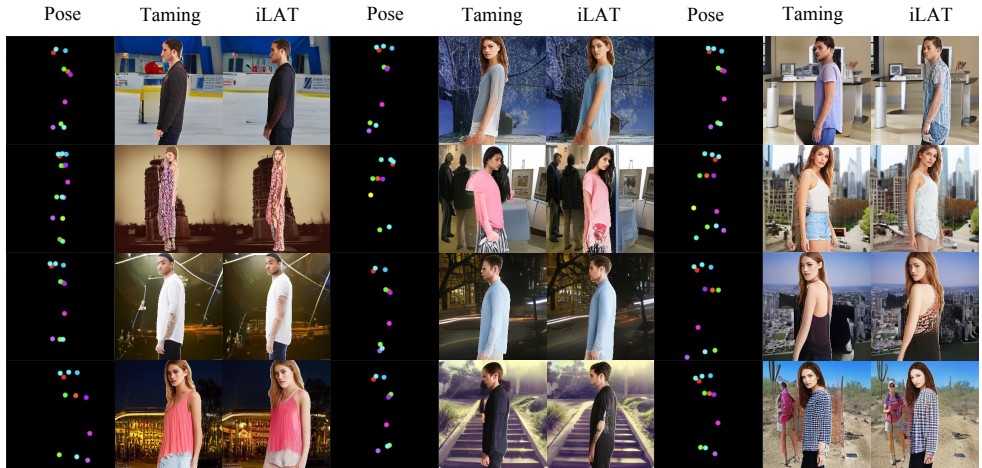

Figure 6: More qualitative results in SDF compared between Taming and iLAT.

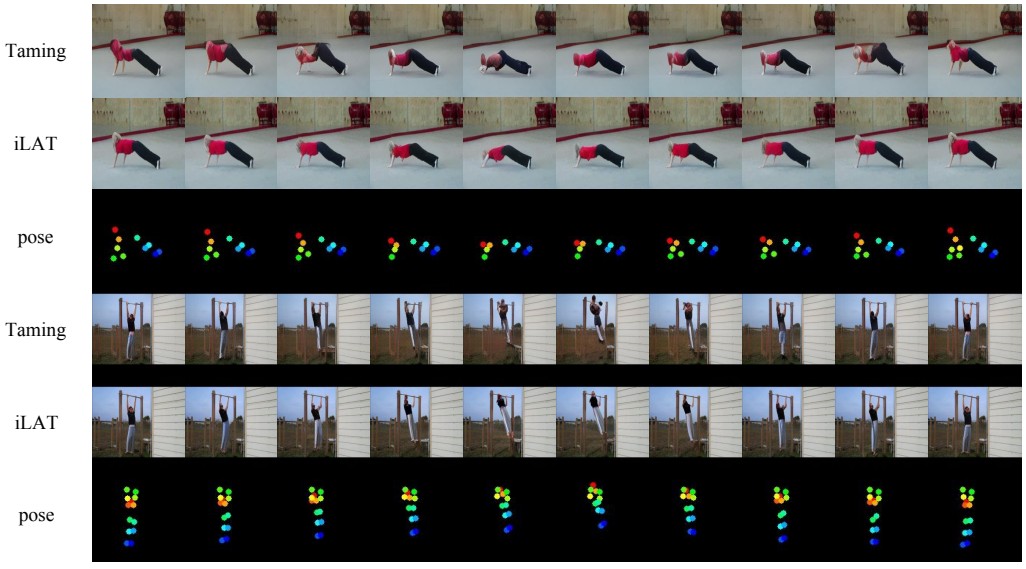

Figure 7: Sequential generation results in PA compared between Taming and iLAT.

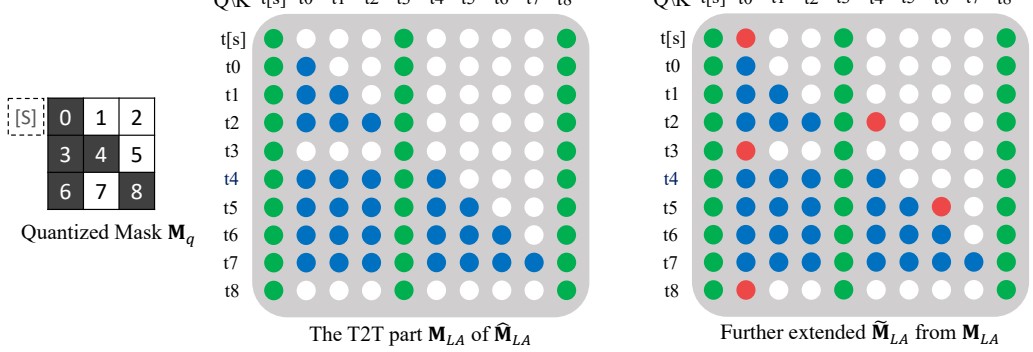

Figure 8: The illustration of the further extended local autoregressive (LA) attention mask $\tilde{\mathbf{M}}_{LA}$, which is hard to achieve parallelly without significant improvements.