# OpenReview forum: "The Image Local Autoregressive Transformer"
_NeurIPS.cc/2021/Conference — NeurIPS 2021 Poster_

### Official Review · Reviewer_BKtm · 2021-07-15

**Rating:** 7
**Confidence:** 3

**Summary:**

This paper focuses on AutoRegressive (AR) models for the image generation empowered by transformers as they show comparable results compared to Generative Adversarial Networks (GANs). Directly applying such AR models to edit/change local image regions, may suffer from the problems of missing global information, slow inference speed, and information leakage of local guidance. This paper focuses on these problems with a novel model – image Local Autoregressive Transformer (iLAT). The proposed method learns the novel local discrete representations, by the newly proposed local lautoregressive (LA) transformer of the attention mask and convolution mechanism. iLAT can efficiently synthesize the local image regions by key guidance and can perform well in various tasks.

**Limitations And Societal Impact:**

Face editing may create some issues such as identity theft so it must be approached carefully.

**Main Review:**

1/ The paper focuses on important problems in AR models such as local generation, semantical consistency, generation of only necessary regions (filling missing regions).

2/ A novel attention mask is proposed to incorporate respective fields of both AR and AE to fit LA generation.

3/ The proposed iLAT dramatically reduces the inference time for local generation, since the only masked region will be generated autoregressively.

4/ Two-stream convolution and a local causal attention mask mechanism are proposed for discrete image encoder and transformer respectively.

5/ The choice of compared works is not clear as there are many GAN-based techniques for missing region completion.

6/ In pose-to-image task, compared to Taming better results are obtained, can authors state the main reasons behind this performance.



**Time Spent Reviewing:**

1 hour

---

> ### Author Response · Authors · 2021-08-09
> **Response to Reviewer 4**
>
> Thank you for the positive feedback regarding our work. We answer your questions below:
>
> Q1: Why compare our method with many GAN-based methods?
>
> Thanks for this point.
>
> (1) GAN-based autoencoder (AE) methods as defined in Line 26-28 are the most popular approaches to solve image editing problems. But these methods show limited performances in non-iconic views (unaligned data with complex scenes). Therefore, we want to indicate that transformer-based image editing is competitive.
>
> (2) Of course, the most important experiment, which is comparing our method with the SOTA transformer-based method --Taming[14], is also shown and discussed in Fig.4, sec.4.1, and sec.4.2 respectively.
>
> Q2: Why can our method achieve better results compared with Taming in pose-to-image tasks?
>
> Thanks for this point.
> For pose-guiding, the main improvement is that our method can avoid generating those complex and difficult non-iconic views with the transformer, especially for backgrounds as mentioned in Line 275-278. Benefits from the decoupling of Eq.4, and the proposed local autoregressive strategy (sec3.2), our transformer can focus on generating masked regions (human poses) with global information efficiently (Line 154-158), and other unmasked regions with complex scenes are recovered by the CNN directly.
>
> Q3: About the societal impact
>
> Thanks for this point. Our method only focuses on technical aspects with open source data. Thus, we will note the potential negative social impacts to the paper.

---

### Official Review · Reviewer_eaP9 · 2021-07-16

**Rating:** 6
**Confidence:** 2

**Summary:**

This paper proposes a transformer-based model for local image editing. The key challenges tackled by the paper is the need to integrate global information, while also preventing leakage of local guidance information (i.e. the local editing should only affect the designated region). To accomplish this, a combination of discrete representation learning (VQVAE) and an autoregresive transformer model is used. The local guidance is enforced through the use of a masked convolution in the feature learning and a local attention mask in the transformer.

**Limitations And Societal Impact:**

The paper is about synthesising realistic media which has many potential negative societal consequences, however no mention of these are made anywhere in the paper. I strongly suggest that a note about the potential negative impacts is added to the paper.

**Main Review:**

**TL;DR:** The paper introduces a useful technique for local image editing / synthesis, but the writing could be substantially improved to make the paper clearer. In the main review below I have labelled points as a strength (+), weakness (-) or mixed (+/-).

**Originality**:

(+) The method proposed in the paper has some novelty. Particularly, I’m not aware of an existing approach that generates local image edits using a transformer model. A transformer does seem ideally suited for this task as they are able to operate on a set of discrete tokens, which means that only the relevant parts of the image can be generated. The proposed components, such as the local attention mask for the transformer and the masked convolution are also to some extent novel, and useful for the task.

(+/-) The authors have described how the work differs from generally related work in a satisfactory manner. However, it would be advisable to add a description of how the proposed method compares to existing transformer-based image synthesis approaches such as [A] and [14] “Taming transformers”. I know [14] is included in the numerical comparison, but it is not clear what parts of the proposed method improve upon [14]. For example, the related work states “recovering images from the discrete codebook still causes blur and artifacts in complex scene” - but the proposed method also uses a discrete codebook. What part of the proposed method addresses this?

[A] Parmar, Niki, et al. "Image transformer." International Conference on Machine Learning. PMLR, 2018.

**Quality**:

(+) The submission appears to be technically sound and I cannot see any major errors or flaws in the method.

(+) The claims made in the paper are largely supported by the experimental results. Specifically, the results show that the proposed approach significantly improves upon the state of the art both qualitatively and quantitatively. The generated images show a significant improvement in detail on both the PA dataset for pose-guided image generation and the face editing task using the FFHQ dataset

(+/-) I would regard the work as complete in terms of the method being well-developed and the convincing results. However, the paper itself can do with some editing as many parts are a bit difficult to read and not expressed very well (see below).

**Clarity**:

(-) The submission is not that clearly written and some explanations are very difficult to follow. For example, at the start of the intro (line 20) the authors state “most works can only handle the images of ‘icon-view’ foreground, rather than the image synthesis of ‘non-iconic view’ foreground”. However, as a reader I have no idea what non-iconic view foreground means. The reference provided ([23] - Microsoft COCO) explains this concept to some extent, but it is by no means a general term. I would suggest the authors add some explanation to the paper to clarify it. Some further parts that need to be clarified:

- What are “respective fields” (ln 50,55,61,310)?

- What does “to lighten the negative influence to the normal CNN learning.” (ln 135) mean?

- What does “condition” and “target” mean in “condition to condition (C2C), condition to target (C2T), target to condition (T2C), and target to target (T2T)”?

- Also, the aspect ratio of some of the figures is severely distorted (especially Figure 4). For a paper about generating high quality images, I would at least expect the basic concepts of image editing to be done correctly.

**Significance**:

(+) In my view the results are quite important as guided image generation is a challenging problem and the proposed approach show a marked improvement over the current methods in the literature. It would, however, be nice to see some higher resolution editing results as the current images seem to be limited to resolutions of 256x256 which is not that useful for real-world use cases.
The idea of editing and synthesising local parts of the image using a transfomer is definitely an idea that others would be keen to build upon. I can see how this could be applied to other problems too (for example in 3D scene and object synthesis).

# Post author response and discussion
After considering the other reviews and the authors' responses, I remain positive about this paper. The authors have promised to address some of the repeated typos and overall, I think this is a good contribution in terms of image editing.The impact of the paper could be improved by some more visually-appealing examples (such as showcasing high-resolution editing), but this is not a fundamental limitation. Therefore, my final rating is "6: Marginally above the acceptance threshold".


**Time Spent Reviewing:**

2

---

> ### Author Response · Authors · 2021-08-09
> **Response to Reviewer 3**
>
> Thank you for your constructive feedback. We answer your questions below:
>
> Q1: Description of the proposed method compared with [A] and [14]
>
> Thanks for suggesting the reference [A]. This is indeed a fundamental and pioneer work that explores the possibility of transformers in the image domain. We will add [A] to the references and discuss it in sec.2, if accepted.
>
> (1) Limited by the main paper space, we summarize all generation methods into autoencoder (AE), autoregressive (AR), and our local autoregressive (LAR) in sec.1 and Fig.1(B). Since Taming [14] can be seen as an AR model, we have compared it with our method in L32-38, and Fig.1(B). Moreover, we have discussed the difference between [14] and our method in Line 102-104 of sec.2.
>
> (2) Model proposed in [A] can be seen as an AR model with local receptive fields. Thus, differences between AR and our LAR have been discussed in Line 32-38, and Fig.1(B).
>
> (3) Built upon the success of [A], we found that Taming [14] proposed in CVPR2021 may potentially have better performance in the task concerned in this paper. Thus Taming could serve as a stronger baseline (than [A]) that we should compare against in a high priority.
>
> Q2: The proposed method also uses a discrete codebook, and how to address the problem of [14]?
>
> Thanks for this question. Since recovering images from the discrete codebook still cause blur and artifacts in complex scenes, we just use discrete codebooks in masked regions as mentioned in Eq.4 and Line 152-156. For complex scenes in unmasked regions, they are recovered by encoder-decoder based CNN directly.
>
> Q3: Minor Questions (1): The concept of ‘non-iconic view’
>
> Thanks for this point. The concept of ‘icon-view’ is discussed in [45]. And authors of [45] cited [23] for it.
> We give the definitions of  ‘icon-view’ and ‘non-iconic view’ in [45] as follows, (a) icon-view: person instances with standing/walking poses near the center of a neatly composed photo in a simple scene. (b) non-iconic view: person instances with arbitrary poses in a cluttered scene.
>
> We will highlight it, if accepted.
>
> Q4: Minor Questions (2):  What are “respective fields” (ln 50,55,61,310)?
>
> Thanks. It is a common concept in the attention mechanism, which also frequently appears in reference [A].
>
> As we know, the final step of self-attention is to aggregate the features according to the attention score with softmax, i.e., one feature is reconstructed by weighted summation from other features. The respective fields mean the valid range that can be attended by every feature, i.e., weights after the softmax>0. The respective fields of the transformer are controlled by attention masks, as mentioned in Line 172-173.
> We also discuss differences among attention masks of AE, AR, and our LAR in Line 194-200.
>
> Q5: Minor Questions (3):  What does “to lighten the negative influence to the normal CNN learning.” (ln 135) mean?
>
> Thanks. As mentioned in Line 60-62, our method solves the ‘information leakage’ caused by normal convolutions with 3*3 or larger kernels (Line 127-132). And we propose the two-stream convolution to tackle this problem as shown in Fig.3(a) through replacing features with leaked information with masked features layer by layer. Thus, the influence of information leakage will be eliminated without hurting the integrities of both masked and unmasked features.
>
> We will improve the presentation, if accepted.
>
> Q6: Minor Questions (4):  What does “condition” and “target” mean in “condition to condition (C2C), condition to target (C2T), target to condition (T2C), and target to target (T2T)”?
>
> (1) condition: source image discrete codebooks
> (pose: source image tokens and pose landmarks vectors, face: sketches tokens)
>
> (2) target: target image discrete codebooks
>
> They have been discussed in sec.3.3 as detailed implementations.
>
> Q7: Minor Questions (5):  the aspect ratio of some of the figures is severely distorted in Fig.4.
>
> Thanks for this point. The aspect ratio of all pictures in Fig.4 is 1:1, which has not been changed. As illustrated in L127-128, for previous CNN-GAN based methods, we follow the experiment settings from [45], which are trained with 128*128, and then resized into 256*256. Since generating ‘non-iconic view’ is very nontrivial for previous methods, these methods even can not achieve acceptable results in 128*128.
>
> Of course, we will carefully improve the layouts of all figures, if accepted.
>
> For the Significance mentioned by the reviewer:
>
> Thanks for approving our method. Exactly, local image editing enjoys more useful real-world applications. Our work has proposed and solved two important problems in local transformer editing, i.e, maintaining global information and preventing information leakage with more efficient inference speed. We will also explore other interesting works as mentioned by the reviewer, e.g, high-resolution, 3D scenes, image inpainting, and object synthesis.
>
> Q8: About the societal impact
>
> Thanks for this point. Our method only focuses on technical aspects with open source data. Thus, we will note the potential negative social impacts to the paper.

---

> > ### Comment · Reviewer_eaP9 · 2021-09-03
> > **Re: response to Reviewer 3**
> >
> > Thanks for the detailed response.
> >
> > In terms of Q4, is the term "respective field" in the paper and reply above not perhaps meant to be "receptive field"? I cannot field anything in [A] or other papers that mention "respective field". If so, I think this and the other writing issues definitely need to be addressed.
> >
> > Overall, I think this is a good paper and my rating remains positive.The results have possibly not been presented in as an eye-catching way as the topic permits, but this is not indicative of the quality of the paper.

---

> > > ### Author Response · Authors · 2021-09-04
> > > **Re: Re: response to Reviewer 3**
> > >
> > > Thanks for your kind and positive advice. Actually, 'respective field' is a typo, and it should be 'receptive field'. We will definitely address it.

---

### Official Review · Reviewer_yqns · 2021-07-16

**Rating:** 4
**Confidence:** 5

**Summary:**

This paper aims to utilize the auto-regressive generative model to achieve the local image editing based on previous state-of-the-art method *taming transformer*.  Specifically, to prevent the information leakage while encoding the input image into discrete sequences, a very straightforward two-stream convolution is proposed. Besides, this paper also presents a new local auto-regressive transformer to benefit from the capability of generative AR model and the global information simultaneously. The experiments show   some improvements over compared methods.

**Limitations And Societal Impact:**

Please see the above mentioned comments.

**Main Review:**



What this paper wants to achieve, i.e. editing any regions of input image while maintaining the original information, is really interesting, but there indeed are some aspects should be explained and clarified.

* The **real effectiveness** of two-stream convolution is not claimed well. In my understanding, the ultimate target of VQ-VAE is to *faithfully reconstruct the input image*, even though there are masks in the input. However, none experiments analyze why the designed convolution is better or could help this point. By contrast, the main effective part of Sec 3.1 may be Equation 4. However, in Line157-Line159, it claims that such simple trick makes AR inference more efficient, which also seems inaccurate. In my opinion, this finetuning trick actually aims to preserve the boundary textures as much as possible, and the efficiency improvements come from the subsequent design of local auto-regressive transformer. Even so, I believe that there are still cases in which the original local contents could not be preserved well while giving arbitrary downsampled masks. This aspect also should be analyzed in details.

* Sec 3.2 introduces the local autoregressive transformer learning, but is it the strict auto-regressive model? To predict next tokens, you need to rely on the conditions of both past tokens and *future* global tokens, which actually breaks the fixed-order decomposed joint distributions of pixels.

* The comparison with *taming transformer* is not fair. You should re-train the baseline, i.e., conditioned on the edited input to regress the final results. For the unmasked region, pick the top-1 token while sampling to reserve the original content.

* Why does this paper mainly focus on editing? The best way to show the improvements of the proposed method is to conduct experiments on image inpainting task. The rests are just the controllable applications of image inpainting.

Based on these considerations, I think currently this submission could not reach the acceptance bar of NeurIPS venue. My rating is rejection now.


**Time Spent Reviewing:**

6 hours

---

> ### Author Response · Authors · 2021-08-09
> **Response to Reviewer 2**
>
> Thank you for your constructive feedback. We answer your questions below:
>
> Q1: In my understanding, the ultimate target of VQ-VAE is to faithfully reconstruct the input image, even though there are masks in the input.
>
> Thanks, but we would politely clarify some misunderstandings from the reviewer. The whole paper is working on an image editing task, rather than an image inpainting task. For the non-mask image regions, both tasks should reconstruct the images, while the image editing task in the masked image region, should synthesize the masked image regions, according to the guidance information of users. More importantly, our task needs to address the problem of ‘information leakage’ (Line 42-47).
>
> Q2: The real effectiveness of two-stream convolution is not claimed well.
>
> Thanks for this point, but we have to clarify the misunderstanding from the reviewer. The real effectiveness is well explained in Line 127-136. We highlight and summarize it as follows. Particularly,  our method not only focuses on ‘editing any regions of input image while maintaining the original information’, but also solves the ‘information leakage’ issue of the VQGAN+transformer based editing. And the two-stream convolution is proposed to solve this problem.
>
> (1)  The ‘Information leakage’ will break it by leaking masked features into unmasked ones, which causes missing detailed local guidance (Line 44-47).  For example, the VQGAN+transformer model with information leakage can not faithfully synthesize double-fold eyelids according to the sketch guidance as shown in Fig.1(A)-(b).
>
> (2) According to our analysis, ‘information leakage’ is caused by the standard convolutions (with 3*3 or larger kernels), which leak masked information to unmasked regions (Line 127-132). Intuitive visualization of leaked features is shown as red circles in Fig.3(a). Thus, our proposed two-stream convolution is the critical component to solve this problem (Line 67-69). The two-stream convolution replaces leaked features with masked features for each layer as shown in Fig.3(a), Eq.2, and Eq.3. This operation can effectively prevent information leakage during the autoregressive training, and protect integrities for both masked and unmasked features.
>
> (3) We have also analyzed the effectiveness of two-stream convolutions in ablation studies of Line 296-303 and Fig.5. And the motivation for the two-stream convolution is also intuitively discussed with examples (Fig.1(A)-(b)), visualization (Fig.3(a)), and description (Line 127-136). We think that the ‘information leakage’ is a potential problem that is easy to be ignored by all VQ-VAE+transformer based local image editing. So it deserves to be explored and studied.
>
> (4) For Eq.4 in sec.3.1, it is just a way to combine the quantized features in masked regions and unquantized features in unmasked regions. Eq.4 can recover better backgrounds and reduce the sequence length that needs to be autoregressive generated. But it can not prevent ‘information leakage’.
>
> Q3: In L157-159, why do we claim that such a simple trick makes AR inference more efficient? The local AR transformer provides efficiency.
>
> Thanks for this point. We give some more insightful discussion about the reviewer's comments.
>
> (1) Since Eq.4 is used to combine the quantized features in masked regions and unquantized features in unmasked regions, it can significantly reduce the token length that needs to be autoregressive generated (Line 157-158). As is widely known to us, autoregressive generation is costly and slow for inferring the whole image sequence. Thus the generated sequence with a shorter length can improve the efficiency of the inference.
>
> (2) During the inference, the local AR transformer only generates the masked part (z_q) of Eq.4 as mentioned in Eq.5. Therefore, Eq.4 will help improve the efficiency of our model which is formulated in Eq. 5. We will better explain this point in Eq. 4 and Eq. 5 if accepted.
>
> Q4: There are cases that original local contents could not be preserved well, which should be analyzed in detail.
>
> Thanks for the careful reading. We had thoroughly analyzed this point in supplementary (Sec3.2, Fig.3, and Fig.4(g)). Generally, such content changes were caused by the quantization loss of the latent codes, a common problem to VQ-VAE related approaches, including Taming. However, we propose a solution to have the best of both worlds.
>
> Particularly,
> (1) This phenomenon is caused by content loss, which is attributed to the vector quantization of the latent codes in VQ-VAE models. Even combining unmasked features from the encoder directly can not handle this problem completely. However, such slight content degradation will not significantly damage the editing results as discussed in our supplementary.
>
> (2) Furthermore, we also present a solution to address this content degradation. Specifically, we can tackle this problem by simply combining outputs and references with masks as discussed in sec.3.2 of the supplementary (widely used in other editing methods). Besides, after the combining, our method still works better than Taming[14] as Taming loses global information and suffers from obvious unnatural color differences in combined boundaries as shown in Fig.3 of the supplementary.
>
> Q5: The proposed iLAT is not a strict AR model.
>
> Thanks. Yes, this paper presents a novel iLAT model, which is not a traditional AR model, but a local autoregressive transformer model specifically designed for the image editing task. The key insight is exactly as suggested by the reviewer that the future global tokens are known in the image editing task, as in Line 165-167 and Eq.5.
>
> (1) Particularly, we also visually compare the key difference between our iLAT model and traditional AR models in the teaser figure of Fig. 1 (B). We give more discussion in the introduction (Line 48-53) and methodology (Line 194-200).
>
> (2) Critically, using global future tokens is important for our iLAT model. Since all global future tokens provide global information to maintain the semantically consistent results.
>
> (3) Accordingly, with the newly introduced global future tokens, the probability likelihood (Eq.5) and the loss function of our iLAT model (Eq.6) are also updated and different from the traditional AR model.
>
> Of course, we will further claim and distinguish them, if accepted.
>
> Q6: Unfair comparison with taming transformer.
>
> Thanks. We would like to clarify the misunderstanding from the reviewer. Of course, we have retrained all methods including Taming. The open source transformer model of Taming can not be directly compared in our task without retraining.
>
> (1) Specifically, authors in [14] did not release taming models trained for guidances, such as sketches for the editing.
>
> (2) For VQGAN, we retrained it for Taming in FFHQ, and finetuned it with the same steps as our TS-VQGAN in pose-guiding tasks based on the open source ImageNet pretrained weights.
>
> (3) (Without retraining for a fair comparison) shown in Line 250 means that Taming* (use our LAT mask) is not further finetuned based on the retrained Taming.
>
> (4) We use top-1 sampling based AR/LAR generation for both Taming and our method.
>
> We will highlight this point, if accepted.
>
> Q7: Why focus on image editing rather than image inpainting?
>
> Thanks. We would like to take the image inpainting as the future work to explore. However, our iLAT model has the novel component (two-stream conv)  to address the problem of  ‘information leakage’ in image editing tasks, as explained in Line 60- 62.

---

### Official Review · Reviewer_je6a · 2021-07-17

**Rating:** 6
**Confidence:** 2

**Summary:**

This paper proposes to use the self-attention mechanism (in particular, Transformer) to enhance image editing. Experiments show promising performance.

**Limitations And Societal Impact:**

Yes

**Main Review:**

This paper proposes to use the self-attention mechanism (in particular, Transformer) to enhance image editing. Experiments show promising performance.

This paper falls out of my research field. I would try my best to offer some comments.

First, the idea of using Transformer, a recently popular module, to enhance the existing problems (e.g. missing global information) is fine. However, this idea seems to have been proposed by the Taming paper (which is cited frequently in this work). Based on this, I am a bit confused of the contribution of this work. In particular, regarding the issues mentioned in the abstract (missing global information, and information leakage of local guidance), how are they solved by the Transformer design, and can both of them be solved by the Transformer design? In particular, the issue of information leakage issue - is it alleviated by random masking? In summary, this paper should mention clearly about the difference between this paper and the Taming paper, and the additional contribution beyond the Taming paper.

Regarding the network design, since I am not an expert, I can only say that everything looks fine, and I will wait for other reviewers' comments for further judgment.

Regarding the experiments, the shown results seem good, but I have two minor questions in Fig 4. (1) In the left part, why is the resolution of some methods significantly different from others? (2) In the right part, why the proposed method change some non-edited contents in the original image (e.g. in the 3rd row, the letter of S in the right border of the image) ?

Overall, this is a well prepared paper. I give a borderline score and will wait for additional information to make the final decision.

**Time Spent Reviewing:**

1

---

> ### Author Response · Authors · 2021-08-09
> **Response to Reviewer 1**
>
> Thank you for your constructive feedback. We answer your questions below:
>
> Q1: How ‘missing global information’ and ‘information leakage’ are solved?
>
> Thanks. Actually, our contributions over Taming have been thoroughly discussed in the introduction and methodology. Particularly, as we discussed in the introduction (Line 32-38, Line 42-47), Taming paper suffers from these two problems, which are well solved by our proposed ‘local autoregressive transformer learning’ and ‘two-steam convolutions’ respectively.
>
> (1) Specifically, we incorporate both autoencoder (AE) and autoregressive (AR) with a novel attention mask mechanism (sec3.2, Fig.3(b)) for the proposed local autoregressive transformer (LAT) (Line 54-58). Therefore, global unmasked tokens in LAT can tackle the missing global information problem with semantic consistency for local editing problems (Line 184-186).
>
> (2) For the ‘information leakage’ as shown in Fig.1(A)-(b), which causes missing editing details (double-fold eyelids), is solved by our specific two-stream convolutions. The reason and solution for it are shown as follows.
>
> (3) Extensive experiments compared against Taming in Fig.4 and Tab.1 also show our advantage over Taming.
>
> Q2: Is the ‘information leakage’ alleviated by random masking?
>
> (1) No, ‘information leakage’ is caused by the standard convolutions (with 3*3 or larger kernels), which leak masked information to unmasked regions (Line 127-132). An intuitive visualization of leaked features is shown as red circles in Fig.3(a). Such leaking makes the model tend to generate results which differ from guidances (Fig.1(A)-(b), Fig.5(B)).
>
> (2) Our solution: To let the masked areas be completely modified by the guidance, we propose the two-stream convolution to the encoder of VQGAN (i.e., TS-VQGAN). The two-stream convolutions replace leaked features with masked features for each layer as shown in Fig.3(a), Eq.2, and Eq.3. This operation can effectively prevent the information leakage during the autoregressive training, and protect integrities for both masked and unmasked features.
>
> Q3: Minor Questions: (1) Why is the resolution of some methods significantly different from others in Fig.4?
>
> Thanks for the careful reading. As illustrated in L127-128, for previous CNN-GAN based methods, we follow the experiment settings from [45], which are trained with 128*128, and then resized into 256*256.
> Critically, it is very nontrivial for these methods to generate the images of ‘non-iconic view’ (i.e., unaligned pose guiding) at the resolution of  256*256.
>
> Q4:  Minor Questions: (2) In the right part of Fig.4, why does the proposed method change some non-edited contents (e.g., the background letter ‘S’)?
>
> Thanks for the careful reading. We had thoroughly analyzed this point in Supplementary (Sec3.2, Fig.3, and Fig.4(g)). Generally, such content changes were caused by the quantization loss of the latent codes, a common problem to VQ-VAE related approaches, including Taming. However, we propose a solution to have the best of both worlds.
>
> Particularly,
> (1) This phenomenon is caused by content loss, which is attributed to the vector quantization of the latent codes in VQ-VAE models. Even combining unmasked features from encoder directly can not handle this problem completely. However, such slightly content degraded will not significantly damage the editing results as discussed in our Supplementary.
>
> (2) Furthermore, we also present a solution to address this content degradation. Specifically, we can tackle this problem by simply combining outputs and references with masks as discussed in sec.3.2 of the supplementary (widely used in other editing methods). Besides, after the combining, our method still works better than Taming[14] as Taming loses global information and suffers from obvious unnatural color differences in combined boundaries as shown in Fig.3 of the supplementary.

---

> > ### Comment · Reviewer_je6a · 2021-08-31
> > **Thanks for the responses**
> >
> > I appreciate the authors' response to addressing my concerns. I choose to keep the original rating that is slightly positive.

---

> > > ### Author Response · Authors · 2021-09-04
> > > **Re: Thanks for the responses**
> > >
> > > Thanks for your kind and positive advice.

---

### Decision · Program_Chairs · 2021-09-27

**Decision:**

Accept (Poster)

**Comment:**

Three reviewers are positive about the paper, while one is more sceptical. As pointed out by the authors, the latter may be due to some confusion about the task addressed: image editing vs. image inpainting.
So the AC follows the advise of the positive reviewers, and decides to accept the paper for publication in Neurips.
The authors are encouraged to work on the paper to improve the clarity, based on the constructive feedback given by the reviewers.